# The D-dimer level predicts the postoperative prognosis in patients with non-small cell lung cancer

**Yuki Shiina**, **Takahiro Nakajima***, **Takayoshi Yamamoto, Kazuhisa Tanaka, Yuichi Sakairi, Hironobu Wada, Hidemi Suzuki, Ichiro Yoshino**

Department of General Thoracic Surgery, Chiba University Graduate School of Medicine, Chiba Japan

* takahiro_nakajima@med.miyazaki-u.ac.jp

## Abstract

### Background

Carcinoma cells often modulate coagulation and fibrinolysis among cancer patients. Plasma dimerized plasmin fragment D (D-dimer) has been reported as a prognostic marker of various types of malignancies, including non-small cell lung cancer (NSCLC). However, the associations between the plasma D-dimer level and peripheral small NSCLC remain unclear.

### Methods

Three hundred and sixty-two patients with NSCLC who underwent radical surgery were retrospectively reviewed. Patients who received anticoagulation therapy before surgery or who lacked preoperative D-dimer data were excluded. The other 235 patients were divided into a high D-dimer (over 1.0 μg/mL) group (HDD group, n = 47) and a normal D-dimer group (NDD group, n = 188) and investigated for their clinical characteristics, computed tomography (CT) findings, pathological findings, and clinical outcomes.

### Results

The mean D-dimer levels was 2.49±2.58 μg/ml in the HDD group and 0.42±0.23 μg/ml in the NDD group. The HDD group was characterized by a predominance of male gender, older age, pure solid appearance on chest CT, vascular invasion in pathology, and a large solid part of the tumor. The HDD group showed a worse overall survival, disease-free survival, and disease-specific survival than the NDD group (p<0.001, <0.001, <0.001, respectively). These survival features were also observed in p-Stage IA disease. There was no marked survival difference when tumors showed ground-glass opacity on CT.

### Conclusion

In NSCLC patients with a solid tumor appearance on CT, high D-dimer levels predict a poor survival and early recurrence.

**Data Availability Statement:** All relevant data are within the manuscript and its Supporting Information files.

**Funding:** NT was supported by JSPS KAKENHI Grant Number JP17K10774.

**Competing interests:** Ichiro Yoshino received honorarium from Pfizer Inc., SHIONOGI&CO., LTD., Astellas Pharma Inc., Nippon Boehringer Ingelheim CO., Ltd., DAIICHI SANKYO COMPANY, LIMITED., Chugai Pharmaceutical Co., Ltd., ONO PHARMACEUTICAL CO., LTD., TEIJIN PHARMA LIMITED. Takahiro Nakajima received honorarium from Olympus Corporation, AstraZeneca plc. This does not alter our adherence to PLOS ONE policies on sharing data and materials.

# Introduction

Carcinoma cells often affect coagulation and fibrinolysis in cancer patients due to their inducing cytokines and coagulation factors. Plasma dimerized plasmin fragment D (D-dimer) has been reported as a prognostic marker of various types of malignancies. Man et al. showed that pretreatment plasma D-dimer, fibrinogen, and platelet levels reflected the prognosis in patients with epithelial ovarian cancer [1]. Several previous studies have reported that, in operable non-small cell lung cancer (NSCLC) patients, the D-dimer levels predict the risk of postoperative early recurrence and a poor prognosis [2, 3]. Little information has been obtained regarding the relationship between the plasma D-dimer level and the detailed clinicopathologic features of NSCLC patients, although multiple overlapping and interacting mechanisms that can explain the increased incidence of thrombosis in patients with malignancies are reported [4–5].

Recently, improvements in and the spread of computed tomography (CT) have increased the chance of detecting small-size peripheral lung cancers. Hattori et al. showed that patients with tumors showing a ground-glass appearance on CT had a better prognosis than those with tumors with a pure solid appearance [6]. Such a radiologic feature is very important when considering surgical management for peripheral small NSCLC, i.e. whether to perform limited resection or conventional lobectomy.

Therefore, in the present study, we investigated the associations between the plasma D-dimer levels and clinicopathologic factors, including the tumor appearances on chest CT.

# Materials and methods

## Patients

A total of 362 patients with NSCLC who underwent radical surgery at Chiba University Hospital between April 2015 and March 2017 were retrospectively reviewed using a prospectively registered database. Patients who had received anticoagulation therapy that affected the D-dimer level before surgery were excluded. In 235 patients, the plasma D-dimer level was measured within 2 months before surgery for a routine checkup of deep vein thrombosis (n = 78) or for an observational study to monitor postoperative thrombus of pulmonary venous stump (UMIN000017528) (n = 157). We divided the patients into 2 groups: a high D-dimer (over 1.0 μg/mL) group (HDD group, n = 47) and a normal D-dimer (less than 1.0 μg/mL) group (NDD group, n = 188) (Fig 1). The cut off value of D-dimer was decided based on the Japanese guidelines, "Guidelines for Diagnosis, Treatment and Prevention of Pulmonary Thromboembolism and Deep Vein Thrombosis" [7]. The 235 subjects of this study and the 119 patients excluded due to a lack of available D-dimer data showed a similar clinicopathologic profile except for age (S1 Table), and their survival curves were almost perfectly superimposed (S1 Fig).

All patients had undergone thin slice CT (1 mm every 1 mm) prior to surgery, and the CT findings were reviewed by all thoracic surgeons and a board-certified radiologist (AN) at the clinical conference. The two groups were compared for their clinical characteristics, tumor appearance on chest CT, histology and histological subtypes and clinical outcomes. In this study, clinical staging was determined based on preoperative findings for CT, fluorodeoxyglucose-positron emission tomography (PET) and magnetic resonance imaging of the head. Lymph nodes greater than 1.0 cm in the short axis by CT or with a standardized uptake value (SUV) of > 2.5 by Pere considered positive and subjected to a biopsy for precise nodal staging. TNM staging was coded according to the International Association for the Study of Lung

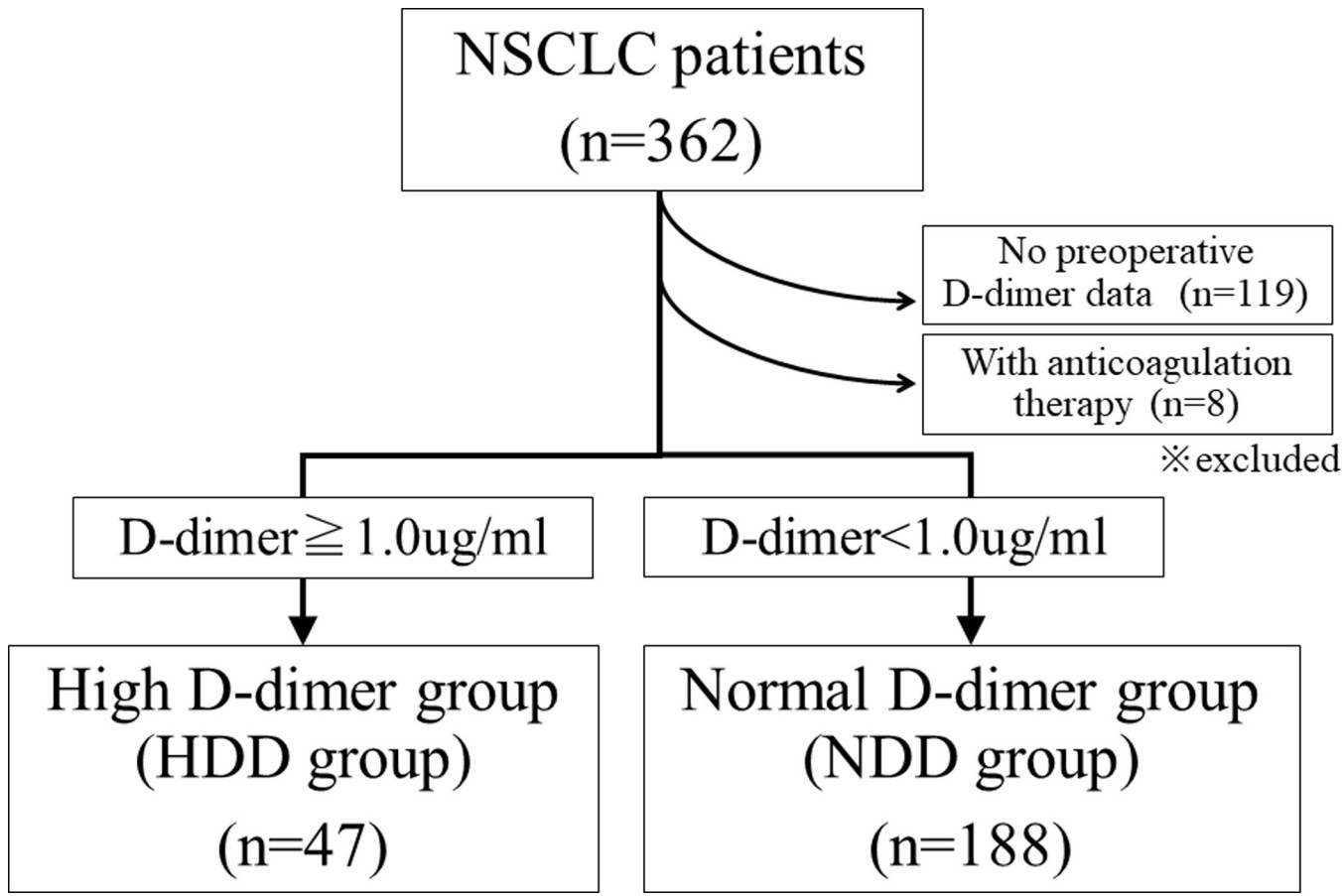

**Fig 1. The flow chart of the patients in this study.** The patients were divided into 2 groups: a high D-dimer (over 1.0 μg/mL) group (HDD group, n = 47) and a normal D-dimer group (NDD group, n = 188).

Cancer (IASLC) staging system (the 8[th] edition). In uni/multiple analyses, not pathological TNM staging but clinical staging is used so as to search for predict survival factors prior to surgery. All patients had 0 or 1 ECOG performance status.

This study was approved by the ethics committee of Chiba University, Graduate School of Medicine (No.3093). All patients' data were fully anonymized before we accessed them and the ethics committee of Chiba University waived the requirement for informed consent.

## Statistical analyses

Statistical analyses were performed using the JMP pro software program, ver. 13 (SAS Institute Inc, Tokyo, Japan). Survival curves were estimated using the Kaplan-Meier method. The log-rank statistic was used for the comparison of the overall survival (OS), disease-free survival (DFS) and disease-specific survival (DSS) distributions. Cox proportional hazards models were used to estimate the hazard ratios for the OS. We considered results to be significant at $p < 0.05$.

## Results

### Patients' characteristics by D-dimer level

The mean D-dimer level was 2.49±2.58 μg/ml in the HDD group and 0.42±0.23 μg/ml in the NDD group. The HDD group was characterized by a predominance of male gender, older age, pure solid appearance on chest CT, a large solid part of the tumor, advanced c-T factor, advanced c-N factor and advanced c-staging compared with the NDD group (Table 1). Regarding pathology and treatment, the HDD group was also characterized by a predominance of advanced p-staging, pathological upstaging, vessel involvement (v+), and undergoing induction therapy than NDD group (Table 2). There were no significant differences in histologic subtypes of adenocarcinomas, surgical procedure, and adjuvant therapy between the two groups.

### Clinical outcomes

During postoperative follow-up, recurrence was observed in 14 (29.8%) and 28 (13.7%) patients in the HDD and NDD groups, respectively (p = 0.015). The HDD group had more distant recurrences than did the NDD group (p = 0.045) (Table 3). In contrast, there was no significant difference in the local recurrence between the 2 groups (p = 0.218). The HDD group showed a worse OS (p<0.001), DFS, (p<0.001) and DSS (p<0.001) than the NDD

**Table 1. Clinical characteristics of both groups.**

| | | HDD group | NDD group | P value |
|---|---|---|---|---|
| | | **n = 47** | **n = 188** | |
| D-dimer | , μg/ml | 2.5±2.6 | 0.4±0.2 | - |
| Age | , years | 71.4±7.5 | 67.0±9.3 | **0.001** |
| Gender | (%) | | | **0.028** |
| | Male | 36 (77) | 110 (59) | |
| | Female | 11 (23) | 78 (41) | |
| Size of the solid part of the tumor (mm) | | 33.3±25.8 | 21.9±15.7 | **0.003** |
| CT appearance | | | | **0.020** |
| | Pure solid | 40 (85) | 123 (65) | |
| | Part solid GGN | 7 (15) | 52 (28) | |
| | Pure GGN | 0 (0) | 13 (7) | |
| cT | | | | **<0.001** |
| | T1 | 24(51) | 138(73) | |
| | T2 | 11(23) | 42(22) | |
| | T3 | 5(11) | 3(2) | |
| | T4 | 7(15) | 5(3) | |
| cN | | | | **0.012** |
| | N0 | 39(83) | 177(94) | |
| | N1-3 | 8(17) | 11(6) | |
| cStage | | | | **0.009** |
| | I | 163(87) | 32(68) | |
| | II | 15(8) | 8(17) | |
| | III | 10(5) | 7(15) | |

D-dimer: plasma dimerized plasmin fragment D, HDD: patients with a high D-dimer level, NDD: patients with a normal D-dimer level, GGN: ground-glass attenuation-dominant nodule, CT: computed tomography

**Table 2. Pathological and treatment parameters of both groups.**

| | | HDD group | NDD group | P value |
|---|---|---|---|---|
| | | n = 47 | n = 188 | |
| p-Stage | | | | 0.398 |
| | I | 27 (57) | 146 (78) | |
| | II | 10 (21) | 25 (13) | |
| | III | 10 (21) | 17 (9) | |
| Pathologic upstaging | | 19(40) | 41(20) | **0.004** |
| Histology | | | | 0.185 |
| | Adenocarcinoma | 31 (66) | 155 (82) | |
| | Squamous cell carcinoma | 14 (30) | 41 (22) | |
| | LCNEC | 0 (0) | 2 (1) | |
| | Large cell carcinoma | 1 (2) | 1 (1) | |
| | Pleomorphic carcinoma | 0 (0) | 2 (1) | |
| | Carcinoid | 0 (0) | 2 (1) | |
| Micropapillary pattern | | | | 0.746 |
| | (+) | 2 (4) | 14 (30) | |
| | (-) | 45 (96) | 174 (93) | |
| pleural invasion | | | | 0.416 |
| | (-) | 40 (85) | 144 (77) | |
| | (+) | 7 (15) | 44 (23) | |
| Nodule count | | | | 0.700 |
| | Single nodule | 46 (98) | 184 (98) | |
| | Separate nodule in the same lobe | 0 (0) | 1 (1) | |
| | Separate nodule in a different ipsilateral lobe | 1 (2) | 1 (1) | |
| Lymphatic invasion | | | | 0.066 |
| | (-) | 38 (81) | 168 (89) | |
| | (+) | 9 (19) | 18 (10) | |
| Vascular invasion | | | | **0.014** |
| | (-) | 28 (60) | 148 (79) | |
| | (+) | 19 (40) | 40 (21) | |
| Surgical procedure | | | | 0.433 |
| | Lobectomy or more | 12(26) | 38(20) | |
| | Sublobar resection | 35(75) | 150(80) | |
| Induction therapy | | 6(13) | 4(2) | **<0.001** |
| | Chemotherapy | 2(4) | 2(1) | |
| | Chemo-radiation therapy | 4(8) | 2(1) | |
| Adjuvant therapy | | 9(19) | 38(19) | 0.984 |
| | Chemotherapy | 6(13) | 36(18) | |
| | Radiation therapy | 1(2) | 1(0) | |
| | Chemo-radiation therapy | 2(4) | 1(0) | |

HDD: patients with a high D-dimer level, NDD: patients with a normal D-dimer level, LCNEC: large cell neuroendocrine carcinoma.

group (Fig 2). Even in p-Stage IA patients, the HDD group showed a worse OS (p<0.001), DFS, (p<0.001) and DSS (p<0.001) than the NDD group (Fig 3).

In a multivariate analysis, a high D-dimer level (hazard ratio [HR] 5.75; 95% confidence interval [CI], 2.12–15.56; p<0.001) and c-T (HR: 4.60, 95% CI, 1.42–14.92; p = 0.011) were independent prognostic factors (Table 4). The HR of CT findings could not be calculated due

**Table 3. Types of recurrence of both groups.**

|  |  | HDD group n = 47 | NDD group n = 188 | P value |
|---|---|---|---|---|
| **Recurrence** |  |  |  |  |
|  | Local | 6(13) | 14(7) | 0.218 |
|  | Distant | 8(17) | 14(7) | **0.045** |

HDD: patients with a high D-dimer level, NDD: patients with a normal D-dimer level.

to no events occurring in patients with ground-glass attenuation-dominant nodules. In an analysis of patients with a pure solid appearance, a high D-dimer level (HR: 5.50; 95% CI, 1.67–11.56; p = 0.003) and c-T (HR: 4.30, 95% CI, 1.54–18.09; p = 0.011) were also found to be independent prognostic factors according to a multivariate analysis (S2 Table). Regarding the patients with ground-glass attenuation-dominant nodules (n = 72), only 1 patient in the NDD group showed recurrence of disease. There was no significant difference in the OS, DFS or DSS between the HDD and NDD groups. In patients with a pure solid appearance on CT (n = 163), the HDD group showed a worse OS (p < 0.001), DFS (p = 0.008) and DSS (p = 0.002) than the NDD group (Fig 4).

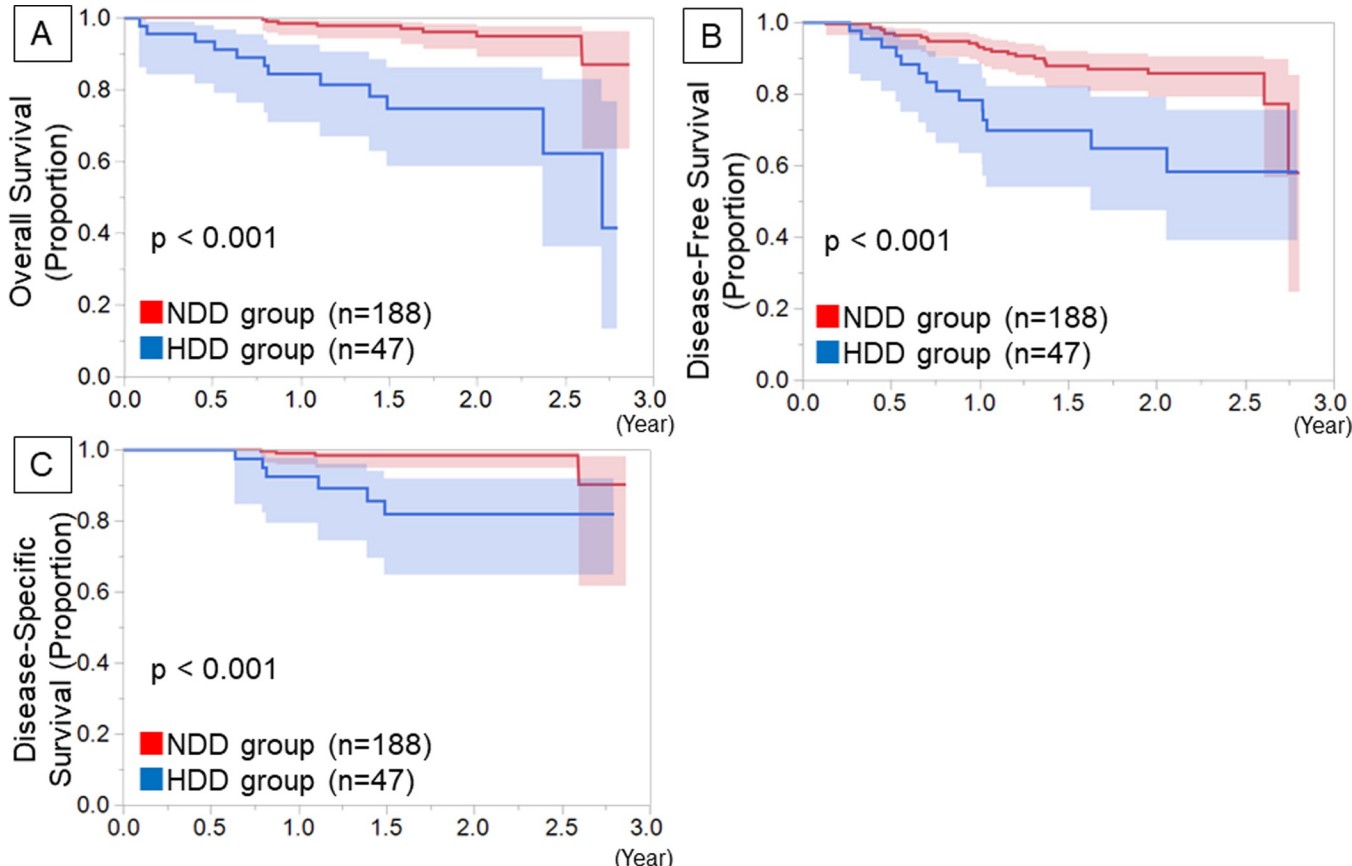

**Fig 2. Kaplan-Meier survival curves among the total patients.** Kaplan-Meier survival curves of the postoperative overall survival (A), disease-free survival (B) and disease-specific survival (C) by preoperative D-dimer level among the total patients in this study.

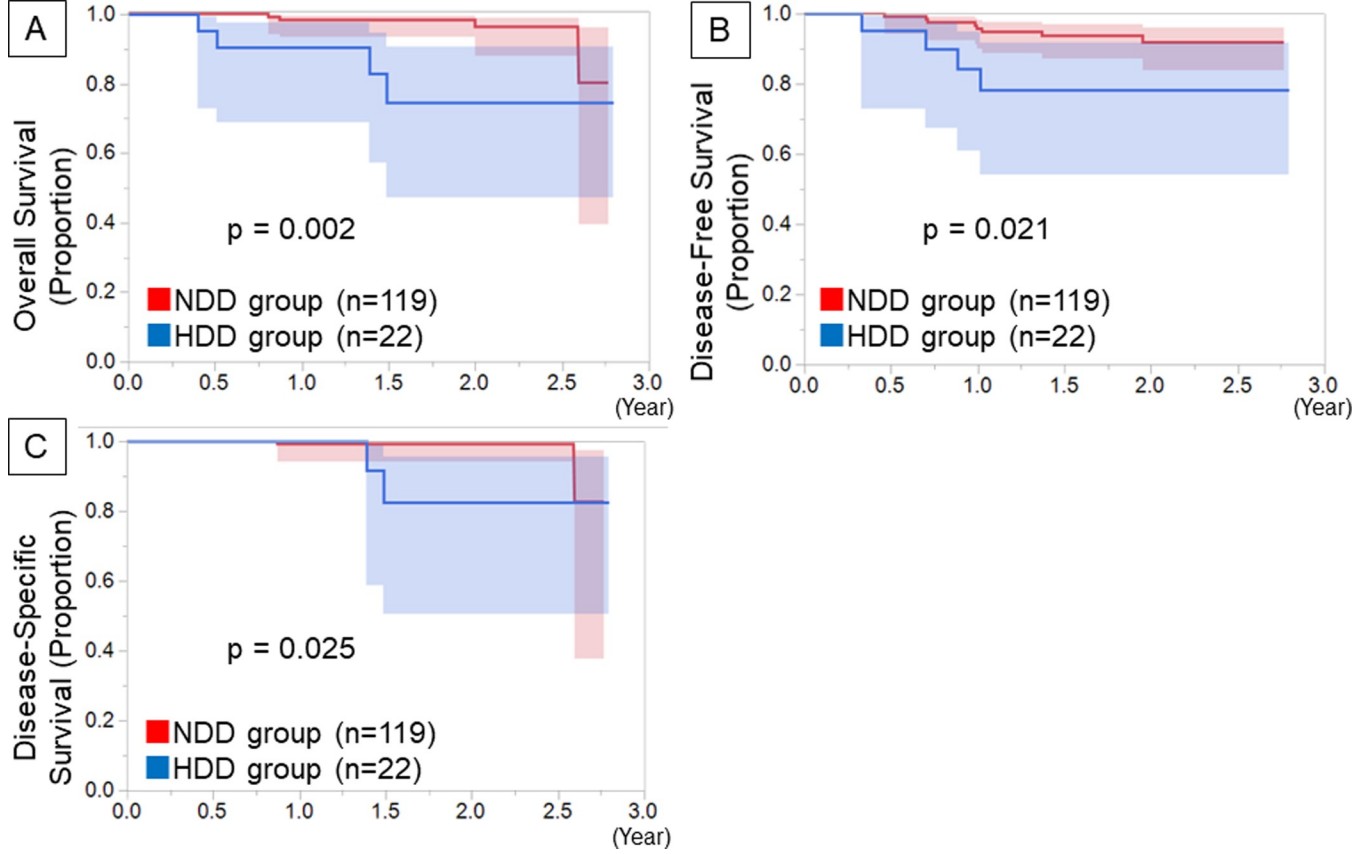

**Fig 3. Kaplan-Meier survival curves among patients with p-Stage IA disease.** Kaplan-Meier survival curves of the postoperative overall survival (A), disease-free survival (B) and disease-specific survival (C) by preoperative D-dimer level among patients with p-Stage IA disease.

## Discussion

Peripheral small-size lung cancer is basically treated by lobectomy if possible, with the role of segmentectomy for such tumors still controversial [8–10]. Even among peripheral early-stage lung cancers, micropapillary- or solid predominant-type adenocarcinoma has been reported to have a poor prognosis [11, 12], and anatomical lobectomy is recommended over segmentectomy because of the increased frequency of local recurrence [13]. NSCLC with ground-glass opacity on CT has shown favor clinical outcomes; however, the preoperative projection of the prognosis is difficult in NSCLC with a pure solid appearance.

We focused on the serum D-dimer level and CT findings to identify new predictors of postoperative outcomes in peripheral small NSCLC. While no correlation was noted between the histology or histological subtypes and the D-dimer level, surgical pathology revealed a relationship between the D-dimer level and vessel involvement of tumors. However, we also found that the patients with ground-glass opacity had favor prognoses despite their serum level of D-dimer. Hattori et al. recently showed that the presence of a ground-glass nodule component is a significant prognostic factor in early-stage NSCLC and that patients with ground-grass opacity tumors had an excellent prognosis (≥90%) irrespective of clinical T factors [6, 14]. Given the results of this study, the preoperative serum D-dimer level did not affect in NSCLC patients with ground-glass opacity tumor.

Previous studies have reported the utility of several coagulation parameters, such as D-dimer, fibrinogen or platelet count, as prognostic markers in patients with several types of

**Table 4. Uni- and multivariate analyses for the OS.**

| | | Univariate analysis | | Multivariate analysis | |
|---|---|---|---|---|---|
| | | Hazard ratio (95%CI) | P value | Hazard ratio (95%CI) | P value |
| Age (years) | | | | | |
| | <70 | - | | | |
| | ≥70 | 1.34 (0.54–3.38) | 0.522 | | |
| Gender | | | | | |
| | Male | 2.34 (0.85–8.21) | 0.105 | | |
| | Female | - | | | |
| cN | | | | | |
| | 0 | - | | - | |
| | 1 | 1.39 (0.08–6.84) | 0.760 | 1.35 (0.07–7.59) | 0.788 |
| | 2 | 3.60 (0.57–12.71) | 0.147 | 2.03 (0.30–8.22) | 0.412 |
| | 3 | - | | - | |
| cT | | | | | |
| | 1 | - | | - | |
| | 2 | 0.89 (0.20–2.91) | 0.857 | 0.69 (0.17–2.86) | 0.604 |
| | 3 | 5.48 (0.84–20.91) | 0.070 | 3.00 (0.63–14.36) | 0.169 |
| | 4 | 9.29 (2.88–26.29) | **< 0.001** | 4.60 (1.42–14.92) | **0.011** |
| CT appearance | | | | | |
| | Pure solid | | - * | | - * |
| | With GGN | | | | |
| D-dimer (μg/mL) | | | | | |
| | ≥1 | 7.54(3.12–19.26) | **< 0.001** | 5.75(2.12–15.56) | **< 0.001** |
| | < 1 | - | | - | |

GGN: ground-glass attenuation-dominant nodule, CI: confidence interval, CT: computed tomography, OS: overall survival

*The hazard ratio of computed tomography findings could not be calculated because no events occurred in patients with ground-glass attenuation-dominant nodules.

malignancies, including lung cancer. Ma et al. showed that high serum D-dimer levels were associated with a poor prognosis in lung cancer patients in a meta-analysis of 11 studies [15]. Zhu et al. showed that D-dimer and fibrinogen could be used to predict the chemotherapy efficacy and prognosis in patients with small cell lung cancer [16]. Inal et al. showed that the D-dimer levels were decreased in chemotherapy responders but increased in non-responders among lung cancer patients [17]. Several recent studies have described the association of a poor prognosis with high D-dimer levels in operable NSCLC patients [2–3]. Gao et al. revealed that the D-dimer level is useful for predicting lymph node metastasis [18]. Several prospective studies of other organs have shown that the association of high coagulation parameters with a poor prognosis was independent of venous thromboembolism [19–20]. The D-dimer level might thus affect the prognosis solely through an important role in tumorigenesis separately from the venous thromboembolism pathway.

Several reports have shown that tumor-mediated coagulation activation is associated with tumor growth, angiogenesis promotion and metastasis [21]. Platelets are reported to increase the metastatic success via multiple mechanisms, including direct shielding of tumor cells and protection of tumor cells from cytokines [22–23]. The fibrinolytic system has been reported to promote tumor growth through several different mechanisms, including angiogenesis, suppressing apoptosis, proliferation of tumor cells and degradation of the extracellular matrix [24]. These previous findings may explain why the HDD group had more distant metastasis in

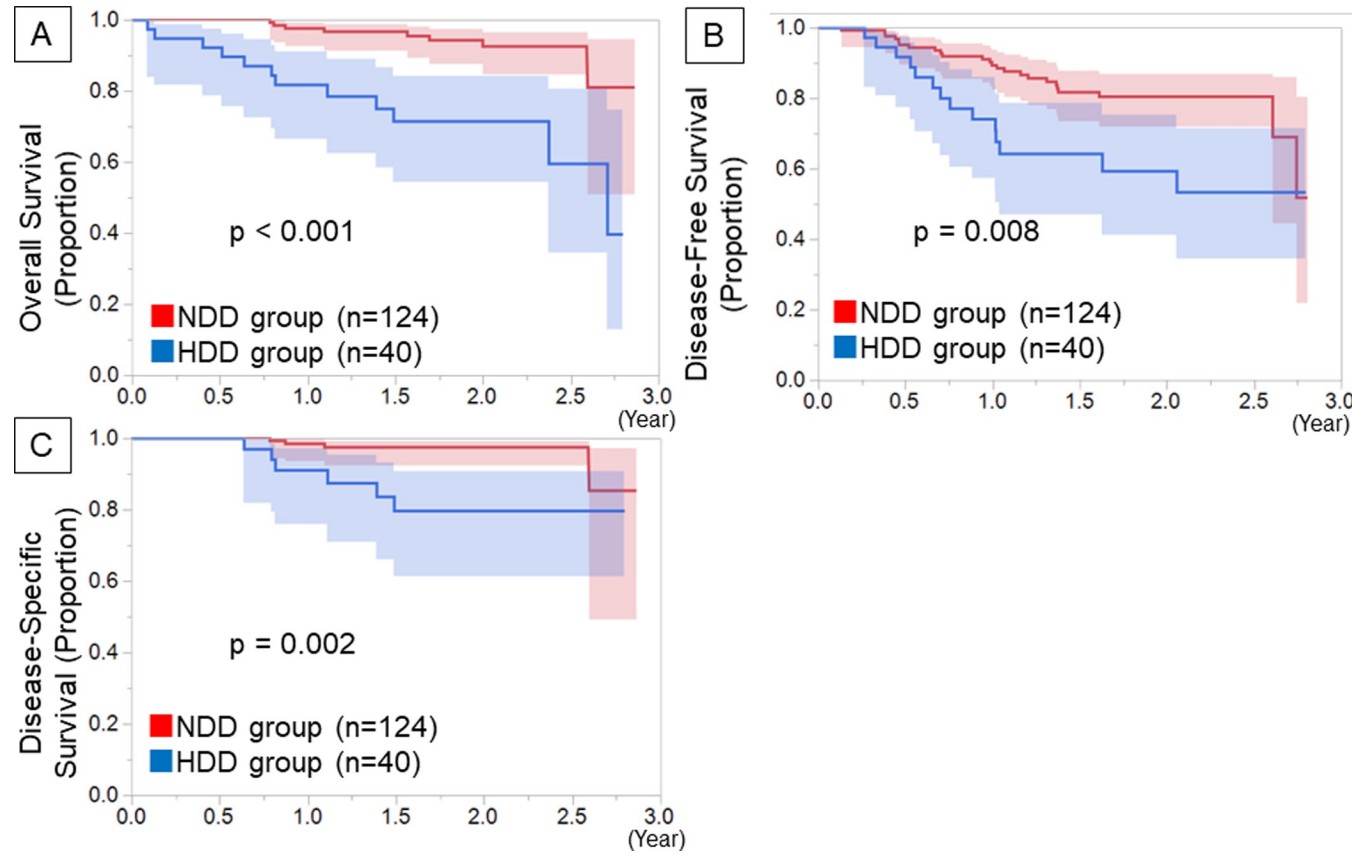

**Fig 4. Kaplan-Meier survival curves among patients with solid CT appearance.** Kaplan-Meier survival curves of postoperative overall survival (A), disease free survival (B), and disease specific survival (C) by preoperative D-dimer level, among patients with solid CT appearance.

the present study than NDD group did. However, the mechanism underlying the association between the serum D-dimer level and the aggressiveness of NSCLC remains unclear. Regarding the pathology in the present study, tumors in the HDD group frequently invaded vessels. Some studies have found that thrombin increased the invasiveness of cancer [25–26], although no report has mentioned the coagulation system and vessel invasion. Although increased D-dimer levels might be a result of vessel injury due to tumor invasion, the association between the D-dimer levels and vessel invasion is unclear and more detailed studies are required.

One limitation of this study was the retrospective nature of the analysis and its performance at a single center. Most of the patients in this study were subjects of an observational study to monitor pulmonary venous thrombosis, as described in the Materials and Methods section; however, there were no marked differences in the clinicopathologic profiles and survival outcomes between the 235 subjects of this study and the 119 patients who were excluded due to a lack of available D-dimer data. A prospective study will be needed to confirm the existence of a relationship between the D-dimer level and tumor aggressiveness as well as the clinical outcome in NSCLC, especially in peripheral small disease. Another limitation was that no aberrant driver genes were assessed in the study. A certain proportion of patients with adenocarcinoma have driver gene alterations. For example, the existence of epidermal growth factor receptor (EGFR) gene mutations and the use of EGFR tyrosine kinase inhibitors contributes to a better survival.

## Conclusions

In NSCLC patients with a solid tumor appearance on CT, high D-dimer levels predict a poor survival and early recurrence. The type of surgery as well as careful post-operative follow-up should be considered in this population.

## Supporting information

**S1 Fig. Survival curves of the subject of this study and excluded patients.** For the 235 subjects of this study and 119 excluded patients due to no available D-dimer data, Kaplan-Meier curves of postoperative overall survival (A), disease free survival (B) and disease specific survival (C) were compared.
(TIF)

**S1 Table. Clinicopathologic profiles of the subjects of this study and patients excluded from the study.**
(DOCX)

**S2 Table. Uni- and multivariate analyses for the OS in patients with a pure solid appearance.**
(DOCX)

**S1 Dataset. The data of patients used in this study.**
(XLS)

## Author Contributions

**Conceptualization:** Takahiro Nakajima.

**Data curation:** Yuki Shiina, Takayoshi Yamamoto, Kazuhisa Tanaka, Yuichi Sakairi, Hironobu Wada, Hidemi Suzuki.

**Formal analysis:** Yuki Shiina.

**Funding acquisition:** Takahiro Nakajima.

**Investigation:** Takahiro Nakajima.

**Project administration:** Ichiro Yoshino.

**Supervision:** Ichiro Yoshino.

**Validation:** Takayoshi Yamamoto, Kazuhisa Tanaka, Yuichi Sakairi, Hironobu Wada, Hidemi Suzuki.

**Writing – original draft:** Yuki Shiina.

**Writing – review & editing:** Takahiro Nakajima, Ichiro Yoshino.

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
