## [Decision Letter · Decision Letter 0]

23 Jul 2019

PONE-D-19-18111

The D-dimer level predicts the postoperative prognosis in patients with non-small cell lung cancer

PLOS ONE

Dear Dr. Nakajima,

Thank you for submitting your manuscript to PLOS ONE. After careful consideration, we feel that it has merit but does not fully meet PLOS ONE’s publication criteria as it currently stands. Therefore, we invite you to submit a revised version of the manuscript that addresses the points raised during the review process.

(1) Description of a normal range of D-dimer

(2) Discussion about therapeutic intervention for patients with non-small cell lung cancer (NSCLC) who underwent radical surgery

(3) Discussion about the effects of surgical procedure, if any

(4) Discussion related to vascular invasion and distant metastasis

(5) Discussion related to venous thromboembolism (VTE) and prognosis

(6) Other issues pointed out by Reviewers

We would appreciate receiving your revised manuscript by Sep 06 2019 11:59PM. To enhance the reproducibility of your results, we recommend that if applicable you deposit your laboratory protocols in protocols.io, where a protocol can be assigned its own identifier (DOI) such that it can be cited independently in the future. For instructions see: http://journals.plos.org/plosone/s/submission-guidelines#loc-laboratory-protocols

We look forward to receiving your revised manuscript.

Kind regards,

Masaru Katoh, M.D., Ph.D.

Academic Editor

PLOS ONE

Journal Requirements:

1. In the ethics statement in the manuscript and in the online submission form, please provide additional information about the patient records used in your retrospective study. Specifically, please ensure that you have discussed whether all data were fully anonymized before you accessed them and/or whether the IRB or ethics committee waived the requirement for informed consent. If patients provided informed written consent to have data from their medical records used in research, please include this information.

2. Thank you for including your competing interests statement; "

Ichiro Yoshino received honorarium from Pfizer Inc., SHIONOGI＆CO., LTD., Astellas Pharma Inc., Nippon Boehringer Ingelheim CO., Ltd., DAIICHI SANKYO COMPANY, LIMITED., Chugai Pharmaceutical Co., Ltd., ONO PHARMACEUTICAL CO., LTD., TEIJIN PHARMA LIMITED.

Takahiro Nakajima received honorarium from Olympus Corporation, AstraZeneca plc.

The author and other co-authors have no conflicts of interest."

Reviewers' comments:

Reviewer's Responses to Questions

**Comments to the Author**

1. Is the manuscript technically sound, and do the data support the conclusions?

Reviewer #1: Yes

Reviewer #2: Yes

2. Has the statistical analysis been performed appropriately and rigorously? 

Reviewer #1: Yes

Reviewer #2: Yes

3. Have the authors made all data underlying the findings in their manuscript fully available?

Reviewer #1: Yes

Reviewer #2: Yes

4. Is the manuscript presented in an intelligible fashion and written in standard English?

Reviewer #1: Yes

Reviewer #2: Yes

5. Review Comments to the Author

Reviewer #1: The authors investigated whether D-dimer was a prognostic impactor for patients with non-small cell lung cancers. Although not that novel, the study was reliable and the analysis was appropriate.

1. What's the normal range of D-dimer in the institution?

2. Why ten patients received induction therapy?

3. Since the small number of patients in each cN group, I wonder if it would be integrated into with or without lymph node metastasis.

4. As for HDD group, any further examination or intervention has been used?

5. Did surgical procedure (Lobectomy vs sublobar resection) impact the prognosis?

Reviewer #2: This is an interesting original work, and the only necessary thing to do is to have more appropriate references, shaping a bit different discussion, and thus making more profound and comprehensive explanations for your findings. Just to mention, vascular invasion should be in this context adequately discussed, as well as the tendency of more distant metastasis in the HDD group than in the NDD group although not significant difference in the type of recurrence, as well as NSCLC with a solid tumor appearance on CT in the context of the poorer prognosis. More recent references should be used - there are several large scaled prospective studies that found that the association of high D-dimer levels with poor prognosis was independent of VTE in hematologic malignancies and solid tumors, which raised the question that D-dimer might be able to affect prognosis through a VTE independent pathway, thus pointing that D-dimer may play an important role in the tumorigenesis.

Some suggested references, among necessary to add and comment:

-Lyman GH, Khorana AA. Cancer, clots and consensus: new understanding of an old problem. J Clin Oncol. 2009; 27:4821-4826.

-Zhao J, Zhao M, Jin B, et al. Tumor response and survival in patients with advanced non-small-cell lung cancer: the predictive value of chemotherapy-induced changes in fibrinogen. BMC Cancer 2012; 12:330.

-Zhu L, Liu B, Zhao Y, Liu L, Yang C, Yang Y, Zhong H. High levels of D-dimer correlated with disease status and poor prognosis of inoperable metastatic colorectal cancer patients treated with bevacizumab. J Cancer Res Ther. 2014; 10:246-251.

-Chen Y, Yu H, Wu C, Li J, Jiao S, Hu Y, Tao H, Wu B, Li A. Prognostic value of plasma D-dimer levels in patients with small-cell lung cancer. Biomed Pharmacother. 2016; 81:210-217.

6. PLOS authors have the option to publish the peer review history of their article (what does this mean?). If published, this will include your full peer review and any attached files.

Reviewer #1: No

Reviewer #2: No

---

## [Author Response · Author response to Decision Letter 0]

11 Aug 2019

Dear editor

We wish to express our appreciation to the Reviewers for their insightful comments, which have helped us to significantly improve the paper. 

Response to Reviewers

Reviewer #1: The authors investigated whether D-dimer was a prognostic impactor for patients with non-small cell lung cancers. Although not that novel, the study was reliable and the analysis was appropriate.

1. What's the normal range of D-dimer in the institution?

Answer: The normal range of D-dimer was less than 1.0µg/ml. We have added the following sentence.

“a normal D-dimer (less than 1.0 µg/mL) group”, to Page 4, Line 70

2. Why ten patients received induction therapy?

Answer: Ten patients had mediastinal lymph node metastasis which was diagnosed as N2, stage IIIA lung cancer. The multidisciplinary team conference which was consisted of thoracic surgeons, respirologists, medical oncologists, and radiologists decided to perform multidisciplinary treatment including induction chemo-radiotherapy following surgery for these patients.

3. Since the small number of patients in each cN group, I wonder if it would be integrated into with or without lymph node metastasis.

Answer: Thank you very much for your suggestion. We have changed the table 1.

4. As for HDD group, any further examination or intervention has been used?

Answer: almost all patients in this study, both for HDD group and NDD group, underwent contrasted CT scan in order to exclude pulmonary artery embolism prior to thoracotomy. 

5. Did surgical procedure (Lobectomy vs sublobar resection) impact the prognosis?

Answer: Thank you very much for your suggestion. There was no prognostic difference between the patients who underwent lobectomy or more and the patients with sublobar resection.

Reviewer #2: This is an interesting original work, and the only necessary thing to do is to have more appropriate references, shaping a bit different discussion, and thus making more profound and comprehensive explanations for your findings. Just to mention, vascular invasion should be in this context adequately discussed, as well as the tendency of more distant metastasis in the HDD group than in the NDD group although not significant difference in the type of recurrence, as well as NSCLC with a solid tumor appearance on CT in the context of the poorer prognosis. More recent references should be used - there are several large scaled prospective studies that found that the association of high D-dimer levels with poor prognosis was independent of VTE in hematologic malignancies and solid tumors, which raised the question that D-dimer might be able to affect prognosis through a VTE independent pathway, thus pointing that D-dimer may play an important role in the tumorigenesis.

Some suggested references, among necessary to add and comment:

-Lyman GH, Khorana AA. Cancer, clots and consensus: new understanding of an old problem. J Clin Oncol. 2009; 27:4821-4826.

-Zhao J, Zhao M, Jin B, et al. Tumor response and survival in patients with advanced non-small-cell lung cancer: the predictive value of chemotherapy-induced changes in fibrinogen. BMC Cancer 2012; 12:330.

-Zhu L, Liu B, Zhao Y, Liu L, Yang C, Yang Y, Zhong H. High levels of D-dimer correlated with disease status and poor prognosis of inoperable metastatic colorectal cancer patients treated with bevacizumab. J Cancer Res Ther. 2014; 10:246-251.

-Chen Y, Yu H, Wu C, Li J, Jiao S, Hu Y, Tao H, Wu B, Li A. Prognostic value of plasma D-dimer levels in patients with small-cell lung cancer. Biomed Pharmacother. 2016; 81:210-217.

Answer: Thank you very much for your very important comments and suggestions. As the reviewer suggested, recent publications suggested the correlation between D-dimer level and poor prognosis. We have added the suggested reference.

We have added the following sentences. 

 “Several prospective studies of other organs have shown that the association of high coagulation parameters with a poor prognosis was independent of venous thromboembolism [19] [20]. The D-dimer level might thus affect the prognosis solely through an important role in tumorigenesis separately from the venous thromboembolism pathway.” , to Page 15, Line 201

“Several reports have shown that tumor-mediated coagulation activation is associated with tumor growth, angiogenesis promotion and metastasis [21]. Platelets are reported to increase the metastatic success via multiple mechanisms, including direct shielding of tumor cells and protection of tumor cells from cytokines [22] [23]. The fibrinolytic system has been reported to promote tumor growth through several different mechanisms, including angiogenesis, suppressing apoptosis, proliferation of tumor cells and degradation of the extracellular matrix [24]. These previous findings may explain why the HDD group had more distant metastasis in the present study than NDD group did.”, to Page 15, Line 206

“Some studies have found that thrombin increased the invasiveness of cancer [25] [26], although no report has mentioned the coagulation system and vessel invasion. Although increased D-dimer levels might be a result of vessel injury due to tumor invasion, the association between the D-dimer levels and vessel invasion is unclear and more detailed studies are required” , to Page 15, Line 216

As the reviewer suggested, we reanalyze the recurrences, dividing the recurrences into local and distance recurrences. We have also added the following sentences, and modified the table 3. 

“The HDD group had more distant recurrences than did the NDD group (p=0.045) (Table 3). In contrast, there was no significant difference in the local recurrence between the 2 groups (p=0.218).”, to Page 10, line 130

---

## [Decision Letter · Decision Letter 1]

21 Aug 2019

The D-dimer level predicts the postoperative prognosis in patients with non-small cell lung cancer

PONE-D-19-18111R1

Dear Dr. Nakajima,

We are pleased to inform you that your manuscript has been judged scientifically suitable for publication and will be formally accepted for publication once it complies with all outstanding technical requirements.

With kind regards,

Masaru Katoh, M.D., Ph.D.

Academic Editor

PLOS ONE

Additional Editor Comments (optional):

Reviewers' comments:

Reviewer's Responses to Questions

**Comments to the Author**

1. If the authors have adequately addressed your comments raised in a previous round of review and you feel that this manuscript is now acceptable for publication, you may indicate that here to bypass the “Comments to the Author” section, enter your conflict of interest statement in the “Confidential to Editor” section, and submit your "Accept" recommendation.

Reviewer #1: All comments have been addressed

2. Is the manuscript technically sound, and do the data support the conclusions?

Reviewer #1: Yes

3. Has the statistical analysis been performed appropriately and rigorously? 

Reviewer #1: Yes

4. Have the authors made all data underlying the findings in their manuscript fully available?

Reviewer #1: Yes

5. Is the manuscript presented in an intelligible fashion and written in standard English?

Reviewer #1: Yes

6. Review Comments to the Author

Reviewer #1: The author has adequately addressed all the comments. Although not that novel, the study is now acceptable for publication.

7. PLOS authors have the option to publish the peer review history of their article (what does this mean?). If published, this will include your full peer review and any attached files.

Reviewer #1: No

---

## [Editor Report · Acceptance letter]

11 Dec 2019

PONE-D-19-18111R1 

The D-dimer level predicts the postoperative prognosis in patients with non-small cell lung cancer 

Dear Dr. Nakajima:

I am pleased to inform you that your manuscript has been deemed suitable for publication in PLOS ONE. Congratulations! Your manuscript is now with our production department. 

With kind regards,

on behalf of

Dr. Masaru Katoh 

Academic Editor

PLOS ONE